# Bioinspired activation strategies for Peano-HASEL artificial muscle

**Zhaozhen Liu**, **Harrison McAleese**, **Andrew Weightman**, **Glen Cooper** *

Department of Mechanical and Aerospace Engineering, Medical Engineering Research Group, University of Manchester, Manchester, United Kingdom

* glen.cooper@manchester.ac.uk

## Abstract

### Background

Human muscles perform many functions during activities of daily living producing a wide range of force outputs, displacements, and velocities. This versatile ability is believed to be associated with muscle activation strategies, such as the number and position of activated motor units within the muscle, as well as the frequency, magnitude and shape of the activation signal. Activation strategies similar to those in the human neuromuscular system could increase the functionality of artificial muscles. Activation in an artificial muscle is the contraction of a single actuator or multiple actuators within the muscle. The number of activated actuators, timing and magnitude of activation (the activation strategy) will enable modulation of the artificial muscles force, displacement and contraction velocity. These activation strategies will mean that an artificial muscle will be able to change its performance to modulate its displacement, length (maximal contractile strain) and velocity for various loading conditions without altering its hardware–making it more versatile in a range of applications or tasks.

This study aims to investigate the effect of activation strategies on the displacement-time response, force-length relationship, and force-velocity relationship of a Peano-hydraulically amplified self-healing electrostatic (HASEL) artificial muscle.

### Method

This study developed a finite element model of an artificial muscle consisting of four Peano-HASEL actuators arranged in three parallel groups in a diamond pattern (two actuators in series in the middle–middle actuators, with one actuator in parallel either side–side actuators). Bioinspired activation strategies were applied to the artificial muscle. Specifically, the number of activated actuators (i.e., activation level), the position of activated actuators, the profile, frequency, and phase of the activation signal were investigated.

### Results

Activating more actuators resulted in increased displacement (106%) and increased average contraction velocity (128%), but overall energy efficiency was sacrificed by 47%. The distortion of inactivated actuators was mitigated by symmetric and phased activation. Phased activation refers to activating middle actuators before side actuators. In addition,

**Data Availability Statement:** All relevant data are within the manuscript and its Supporting Information files. Specifically I have included a compressed file S2 with the FE model file and all

data files in txt format which have been used in the figures. Additionally there is also a contents file to describe what the files are.

**Funding:** The author(s) received no specific funding for this work.

displacement patterns of the Peano-HASEL artificial muscle changed with activation signal frequency. The ramp activation signal with low frequencies (less than 5 Hz) is suitable for applications favouring controllable displacement, while the step activation signal produces greater average contraction velocity (325%) which would be advantageous for applications requiring a fast response.

## Conclusion

This paper demonstrates that activation strategies can enhance multi-actuator artificial muscle function without changing the physical hardware configuration. Specifically, activation strategy can, improve displacement control, contraction velocity and output force. Future work should focus on more complex artificial muscle arrangements and test activation strategies in practical experiments.

## Introduction

Artificial muscles, known as muscle-like actuators, refer to materials or devices that can reversibly deform due to an external stimulus and are in high demand in robotic and medical applications [1]. Peano-hydraulically amplified self-healing electrostatic (HASEL) actuators are a promising electroactive polymer-based artificial muscle, which have similar stress output (0.3 $MPa$), work density (64 $kJ/m^3$) and specific energy (0.6 $kW/kg$) to biological muscle [1,2].

Some progress has been made in the implementation of the Peano-HASEL actuators (PH actuators) in practical applications, including the optimization of force characteristics [3] and the latest understanding of dynamic behaviour of the actuators [4]. The force-length (i.e., maximum contractile strain under applied loads) and force-velocity relationships of PH actuators have been optimized to match the shape of human skeletal muscle fibres [3]. However, the force output of a single PH actuator is insufficient to meet the needs of many applications such as prosthetics. For instance, the maximum force output of a single PH actuator with commonly used geometry ranges from 15 to 40 $N$ [2,3], whereas a human skeletal muscle of similar geometry produces forces greater than 200 $N$ [5]. Therefore, multiple PH actuators must be stacked to achieve force amplification. This stacked parallel configuration has been used to create linear grippers [6] and high-speed prosthetic fingers [7]. A bipennate PH artificial muscle consisting of multiple actuators was recently proposed to amplify the force output and restore the force output of the PH artificial muscle to the level of human skeletal muscle. Activation strategies for the PH artificial muscles need to be carefully considered due to the interactions between adjacent PH actuators when activating the PH artificial muscle.

In the human body, the activation strategies of motor units regulate the muscle force development [8]. A motor unit is the basic functional element in the neuromuscular system. The force exerted by a muscle depends on which motor units are recruited and the rate at which they discharge action potentials [9]. Specifically, the more motor units are activated, the higher the exerted force. The spatial distribution and position of activated motor units may also affect the overall displacement and force output [10]. The exerted force of a single motor unit varies from 3 to 15 times for different discharging rates [9]. To contribute smooth and coordinated movement, human muscle must activate the appropriate number and combination of motor units [11,12].

The activation strategies and muscle architecture from the human neuromuscular system have been incorporated into actuator design. The force output of actuators is often amplified

by stacking multiple actuators [3,7,13]. The activation signal for an actuator was optimized in terms of phase and frequency to achieve precise and smooth force output [14]. The calf muscle activation profile was simulated and implemented in a moving vehicle for balance control [15]. Based on the above literature which use biomimetic principles, incorporating muscle activation strategies into PH artificial muscle may be beneficial for damping characteristics and force output profile.

This paper aims to investigate how the activation strategies of PH actuator based artificial muscle (i.e., groups of PH actuators) affect its force output and length, as well force-velocity relationships. Previously researchers have focussed on the physical hardware of artificial muscle design [2,3]. We are investigating their control through activation strategy which may enable additional functional modulation without hardware changes. This could both improve performance but also make artificial muscles more versatile to perform in a range of applications. Other researchers have looked at activation signals but mainly on ramp and magnitude variation [3,6]. Here we present a biomimetic approach to explore the design space in more aspects which we hypothesise will unlock greater functionality. Specifically, activation strategies will be investigated in three aspects: number of activated actuators, position of activated actuators, and activation signals. For the activation signals, the profile, phase, and frequency of the signals will be studied.

## Materials and methods

### A. Finite element modelling of PH artificial muscle

Human skeletal muscle is a highly organized tissue composed of bundles of muscle fibres which are arranged in a complex series and parallel configuration [16]. Similarly, PH actuators

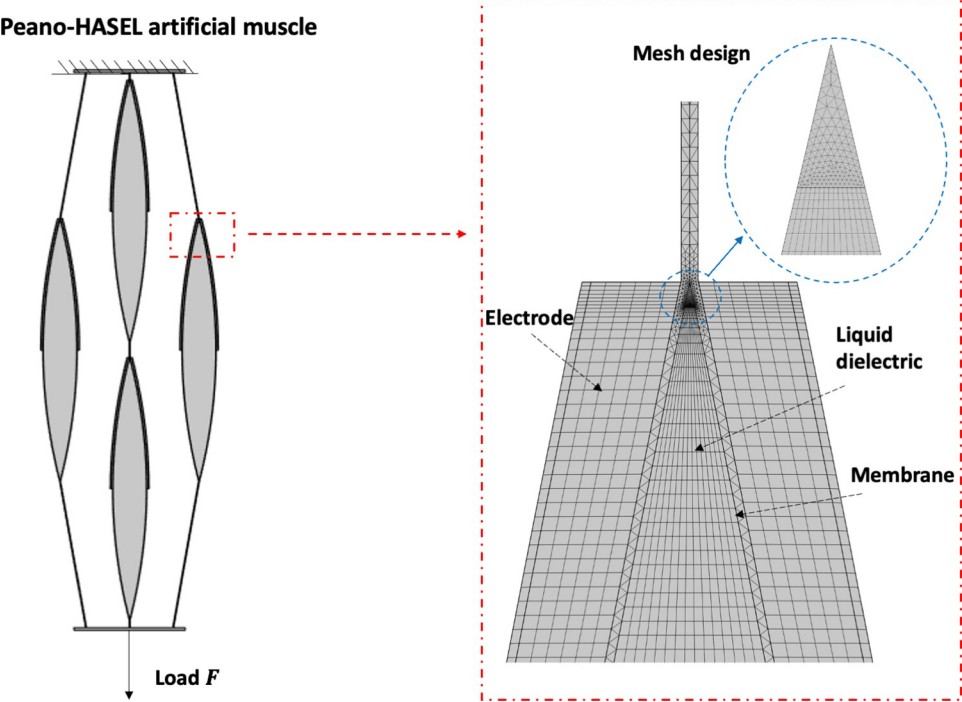

**Fig 1. Finite element model geometry, mesh design and boundary conditions of the PH artificial muscle consisting of four PH actuators.** The top end of the PH artificial muscle was fixed and a range of constant loads, F (0-18N), were applied at the bottom end. Linear triangular, linear mapped, and linear triangular-mapped hybrid elements were used to mesh the membrane, electrode, and liquid dielectric, respectively.

are often stacked in parallel to achieve force amplification [3,6]. The 2D PH artificial muscle model in the present paper is shown in Fig 1 and consists of four actuators connected both in parallel and series to study the effect of these arrangements. The finite element model was built in COMSOL Multiphysics 6.1 (Cambridge, UK). Details of the model are as follows:

**Geometry, materials, and mesh design.**   The geometry of the PH actuators in the computer model was scaled based on the previous geometry commonly used in PH actuator experiments and simulations [3–7]. The out-of-plane depth $w$ was set to 0.04 $m$. The length of the actuator $Lp$ was 0.01 $m$. The length of the electrode $Le$ was 0.005 $m$. The fluid fill volume of one single PH actuator was 1.27 $ml$, corresponding to an initial central angle $\alpha_0$ of 0.24 rad. Materials used for the electrode, the membrane, and the fluid dielectric were graphene ink, biaxially oriented polypropylene (BOPP), and FR3 oil (properties shown in Table 1). The thickness of the BOPP membrane is 28 $\mu m$. The connection between actuators is an extension of the actuator membrane. Therefore, the connections have the same thickness (28 $\mu m$), out-of-plane depth (0.04 $m$), and material properties (BOPP) as the actuator membrane. The length of the connection is approximately half the length of PH actuator, i.e., 0.005 $m$.

There were 37610 domain elements and 14038 boundary elements in the model. Linear triangular, linear mapped, and linear triangular-mapped hybrid elements are used for the membrane, the electrode, and the liquid dielectric, respectively (see Fig 1). The Winslow smoothing method was added to the liquid domain. Mesh refinement was achieved by globally increasing the mesh density until convergence was achieved.

**Boundary conditions.**   The left electrode of each PH actuator was set to a terminal voltage, and the right electrode was grounded. A maximum activation voltage of 6 $kV$ was used in this paper, which was the highest practical voltage that did not cause an arc between wire connections from previous experiments [17]. The top end of the PH artificial muscle was fixed. Loads from 0 $N$ to 18 $N$ were applied at the bottom end in the arrow direction (see Fig 1). Adjacent actuators were initially set not to be in contact with each other which allowed space for their deformation.

The contacts between membranes of PH actuators and between the membrane and the electrode are regarded as non-adhesive contact. The classic Hertz theory is used here to describe the contact behaviour with two main parameters, i.e., contact force $F$ and penetration $\delta$.

The relationship between the contact radius $a$ and penetration $\delta$ is:

$$a = \sqrt{\delta \cdot \frac{L_p - L_e}{\alpha}} \tag{1}$$

Where $L_p$ is the length of the actuator, $L_e$ is the length of the electrode, $\alpha$ is the central angle of the actuator.

**Table 1. Material properties of the Peano-HASEL artificial muscle.**

|  | Material | Young's modulus $E$ ($Pa$) | Poisson's Ratio $v$ | Relative permittivity | Density ($Kg/m^3$) | Dynamic Viscosity ($Pa \cdot s$) |
|---|---|---|---|---|---|---|
| Electrode | Graphene ink | $1.6 \times 10^5$ | 0.3 | - | 1100 | - |
| Membrane | BOPP* | $2.5 \times 10^9$ | 0.33 | 2.2 | 910 | - |
| Liquid dielectric | FR3 Oil | - | - | 3.2 | 960 | 0.06 |

*BOPP-Biaxially-oriented polypropylene.

The relationship between contact force $F$ and the contact radius $\alpha$ is:

$$F = \frac{4E^*a^3}{3(L_p - L_e)/\alpha} \tag{2}$$

$$\frac{1}{E^*} = \frac{1 - v_m^2}{E_m} + \frac{1 - v_e^2}{E_e} \tag{3}$$

Where $E^*$ is reduced modulus, $E_m$ is the young's modulus of the membrane, $E_e$ is the young's modulus of the electrode, $v_m$ is the Poisson ratio of the membrane, and $v_e$ is the Poisson ratio of the electrode.

**Postprocessing.** The displacement-time response under different load conditions (i.e., from no-load condition to maximum load condition) were recorded for the PH artificial muscle. The gradient of the rise time was used to calculate the contraction velocity. Force-length relationship and force-velocity relationship were then plotted based on the results from the model. Force is the load F applied to the PH artificial muscle. Length is defined as the maximum contractile strain under the varying applied load F.

**Validation.** The actuators within the finite element model were compared with previously published results of both an analytical model and an experiment [3] and found to be in close agreement with errors of less than 5%. Additionally in our previous work [17] we compared the finite element model results to experimental results of a bipennate PH actuator artificial muscle (i.e., two PH actuators in a symmetric angled arrangement). This again had low errors of less than 10% across length changes of 1–6%. These finite element models were then extended to create an artificial muscle of four PH actuators in a diamond arrangement, as shown in Fig 1. The diagrams of actuator arrangements and their force-length characteristics compared to experimental and analytical data from the literature can be found in the supplement material S1 Fig.

## B. Activation strategy of PH artificial muscle

Human muscles can achieve a variety of functions during locomotion and activities of daily living, such as isometric, isotonic, and concentric contraction for a large range of force outputs, displacements, and velocities [18]. This ability is believed to be related to muscle activation strategies [19]. Activation strategies similar to those in the human neuromuscular system could enable increased functionality for artificial muscles. Bioinspired activation strategies will be applied to a PH artificial muscle to investigate its effect on force production, displacement, and velocity. Three aspects will be investigated: (i) number of activated actuators, (ii) position of activated actuators, and (iii) activation signals. For activation signals, the frequency and phase of the signals will be studied. The effect of the activation strategies on the performance of the PH artificial muscle was assessed in terms of displacement-time response, force-length relationship, and force-velocity relationship.

One to four actuators were activated in corresponding to 25% to 100% activation level to investigate the effect of the number of activated actuators on the performance of the PH artificial muscle, as shown in Fig 2A. Actuators in orange refers to activated actuators and actuators in grey refers to inactivated actuators. Fig 2B shows the configurations for the study of activated position. For 50% and 75% activation level, the middle actuators (i.e., actuators in the inner layer of the PH artificial muscle) and side actuators (i.e., actuators in the outer layer) were activated in different combinations to study the effect of activation position. Fig 2C shows the settings for the activation signals. The study for activation signals was performed at 50% activation level. A step input and two ramp inputs (with slopes of 240 $V/ms$ and 120 $V/ms$, respectively) were used to activate the actuators to study the effect of signal profile;

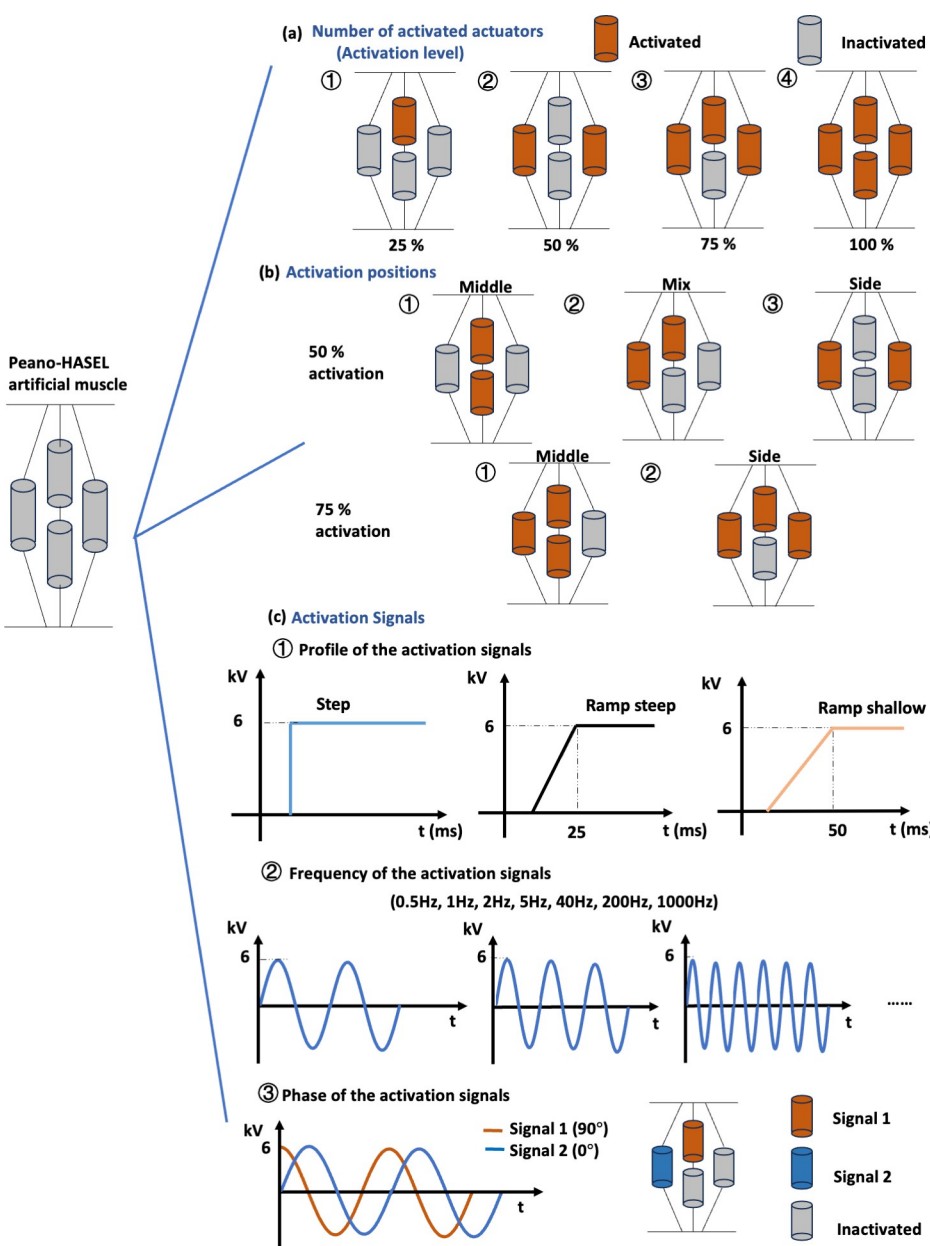

**Fig 2. Activation strategy investigation for the PH artificial muscle consisting of four actuators.** (a). The number of activated actuators was studied. One to four actuators were activated, corresponding to 25% to 100% activation level. (b). The number of activated actuators was studied. For 50% and 75% activation level, the middle and side actuators were activated in different combinations to study the effect of activation position. (c). For the activation signal, the profile, frequency, and phase of the signal were studied. Under 50% activation level, a step input and two ramp inputs were used to activate the actuators to study the effect of signal profile, sinusoidal signals with frequencies ranging from 0.5 $Hz$ to 1000 $Hz$ were used to activate the actuators to study the effect of signal frequency, a sinusoidal signal with a frequency of 40 $Hz$ and a phase of 90 degrees and 0 degree were used to activate the actuators to study the influence of signal phase.

sinusoidal signals with frequencies of 0.5 $Hz$, 1 $Hz$, 2 $Hz$, 5 $Hz$, 40 $Hz$, 200 $Hz$, 1000 $Hz$ were used to activate the actuators to study the effect of signal frequency; a sinusoidal signal with a frequency of 40 $Hz$ and a phase of 90 degrees and 0 degree were used to activate the actuators to study the influence of signal phase.

# Results

## A. Number of activated actuators

Fig 3 shows the results obtained from the finite element model for number of activated actuators. Fig 3A shows the displacement-time responses under no-load condition of four configurations. The more actuators that are activated at the same time, the greater the vibration of the displacement output. Fig 3B and 3C shows the force-length (i.e., maximum contractile strain under applied loads) and force-velocity relationships of four configurations, respectively. For a given load, the more actuators activated, the greater the output strain and contraction velocity. Compared to activating one actuator, activating four actuators results in an average increase of

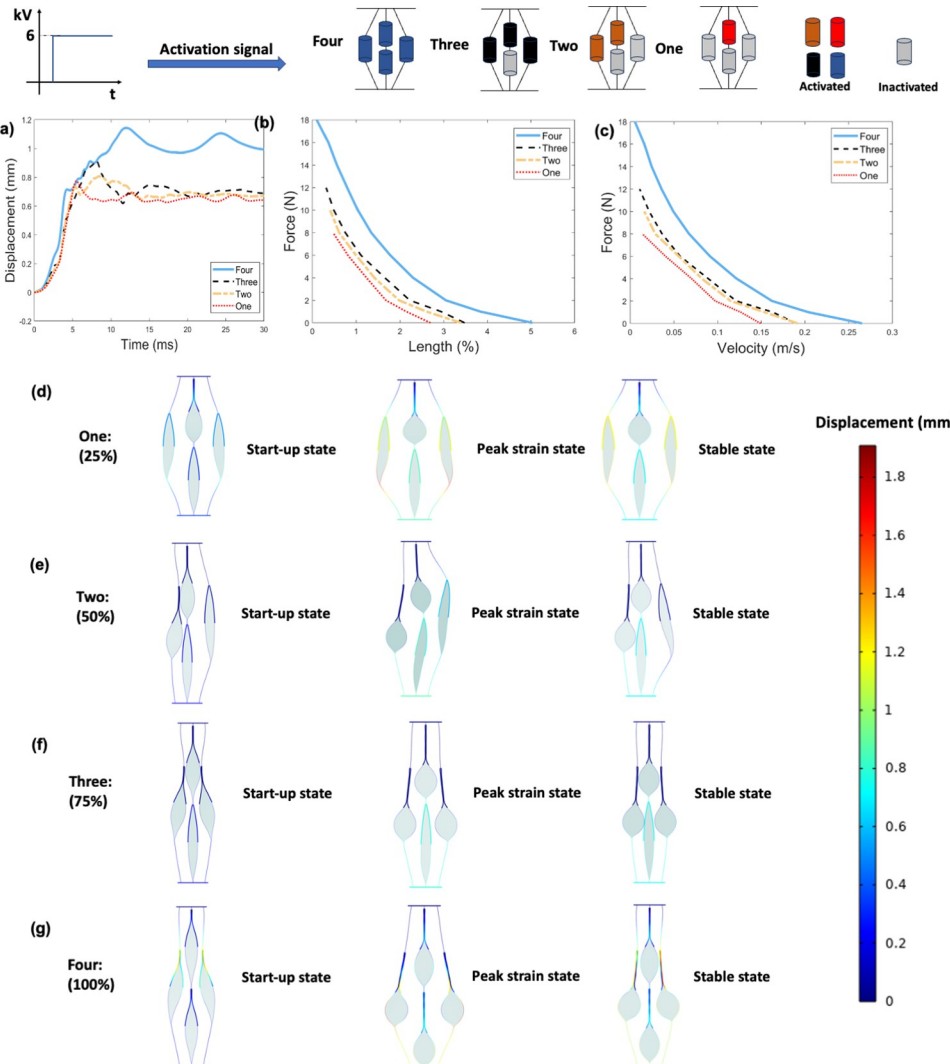

**Fig 3. Results obtained from the finite element model for number of activated actuators.** (a). Displacement-time response under no-load condition. (b). Force-length (i.e., maximum contractile strain under applied loads) relationship (c). Force-velocity relationship. (d). Three states of 25% activation level under no load conditions. Three states are the activated actuator starting to response, the actuator producing the peak strain output, and the actuator reaching a stable position. Colour legend represents the magnitude of displacement. The inside liquid dielectric is marked in pewter grey. (e). Three states of 50% activation level under no load conditions. (f). Three states of 75% activation level under no load conditions. (g). Three states of 100% activation level under no load conditions. Note that activation displacements shown in d-g are 1:1 scale.

106% and 128% in output strain and contraction velocity, respectively. The maximum force output of four actuators is 2.1 times that of one actuator. Considering that activating four actuators requires four times the energy input than activating one actuator, activating all the four actuators leads to a deduction of 47% in efficiency. There is a trade-off between system output and energy efficiency. Muscle activity shows a similar trend in the human body. Human skeletal muscle activation levels vary with different tasks. For example, the neuromuscular system undergoes a reduction in the muscle activation level during long-distance running, indicating that muscle activation strategy adapts to make the system more efficient [20]. Fig 3D to 3G show three states of different activation levels under no load conditions. The colour represents the magnitude of displacement. The activated actuators pull inactivated actuators to move, causing inactivated actuators to distort, which hinders the strain output and contraction velocity.

## B. Position of activated actuators

Fig 4 shows the results obtained from the finite element model for position of activated actuators at 50% activation level. Fig 4A shows the displacement-time responses under no-load condition of three configurations. The middle configuration provides the greatest displacement

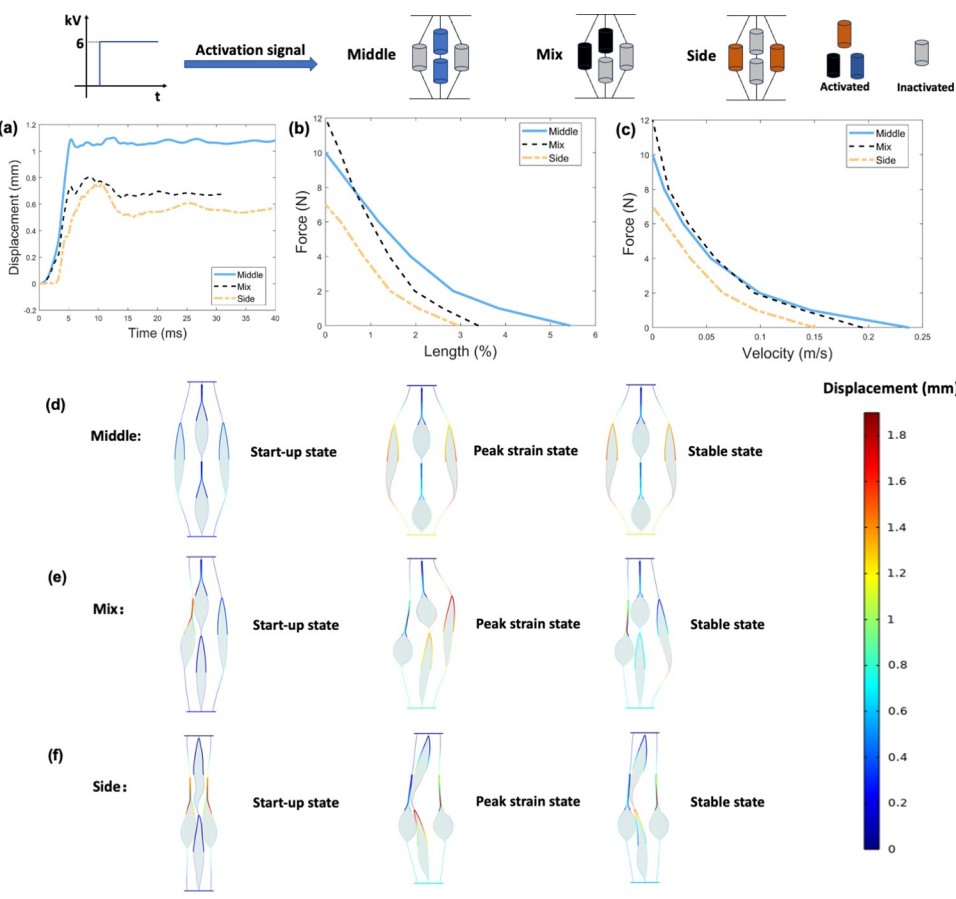

**Fig 4. Results obtained from the finite element model for position of activated actuators at 50% activation level.** (a). Displacement-time response for three configurations under no-load condition. (b). Force-length (i.e., maximum contractile strain under applied loads) relationship (c). Force-velocity relationship. (d). Three states of the middle configuration under no load conditions. (e). Three states of the mix configuration under no load conditions. (f). Three states of the side configuration under no load conditions. Note that activation displacements shown in d-f are 1:1 scale.

output with smallest vibration. Fig 4B and 4C shows the force-length (i.e., maximum contractile strain under applied loads) and force-velocity relationships of three configurations, respectively. For a given load, the more middle actuators that are activated, the greater the output strain and contraction velocity. Compared to activating two side actuators, activating two middle actuators results in an average increase of 130% and 71% in output strain and contraction velocity, respectively. The mix configuration contributes to the greatest force output. The maximum force output of the mix configuration is 170% of that of side configuration. Fig 4D to 4F show three states of different position configurations under no load conditions. Activating side actuators leads to greater distortion of inactivated actuators than activating middle actuators.

Fig 5 shows the results obtained from the finite element model for position of activated actuators at 75% activation level. Fig 5A shows the displacement-time responses under no-load condition of two configurations. Similar to 50% activation level, middle configuration provides the greatest displacement output with smallest vibration. Fig 5B and 5C shows the force-length (i.e., maximum contractile strain under applied loads) and force-velocity relationships of two configurations, respectively. For a given load, the more middle actuators that are activated, the greater the output strain and contraction velocity. Compared to side configuration, middle configuration results in an average increase of 34% and 10% in output strain and contraction velocity, respectively. However, the maximum force output of the side configuration is 114% of that of middle configuration. Fig 5D to 5E show three states of different

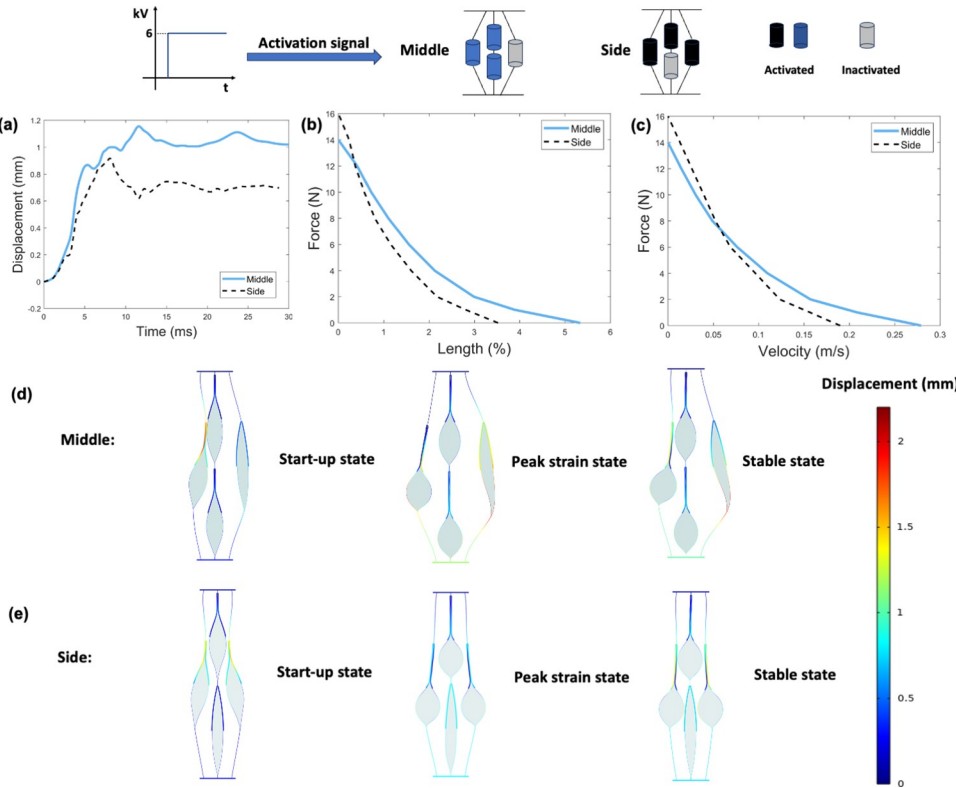

**Fig 5. Results obtained from the finite element model for position of activated actuators at 75% activation level.**
(a). Displacement-time response for two configurations under no-load condition. (b). Force-length (i.e., maximum contractile strain under applied loads) relationship (c). Force-velocity relationship. (d). Three states of the middle configuration under no load conditions. (e). Three states of the side configuration under no load conditions. Note that activation displacements shown in d-e are 1:1 scale.

position configurations under no load conditions. Asymmetric activation leads to greater distortion of inactivated actuators than symmetric activation.

## C. Profile of the activation signal

Fig 6 shows the results obtained from the finite element model for profile of activation signals at 50% activation level. Fig 6A shows the displacement-time responses under no-load condition of three configurations. Displacement output of the step input shows vibration while the two ramp inputs are smooth and have overdamped characteristics. Fig 6B and 6C shows the force-length (i.e., maximum contractile strain under applied loads) and force-velocity relationships of three configurations, respectively. For a given load, the three input signals produce the same level of strain output, while for the contraction velocity, the step input produces greatest contraction velocity. A shallow ramp signal produces a smaller contraction velocity than that of steep ramp signal. Compared to a shallow ramp signal, the step signal results in an average

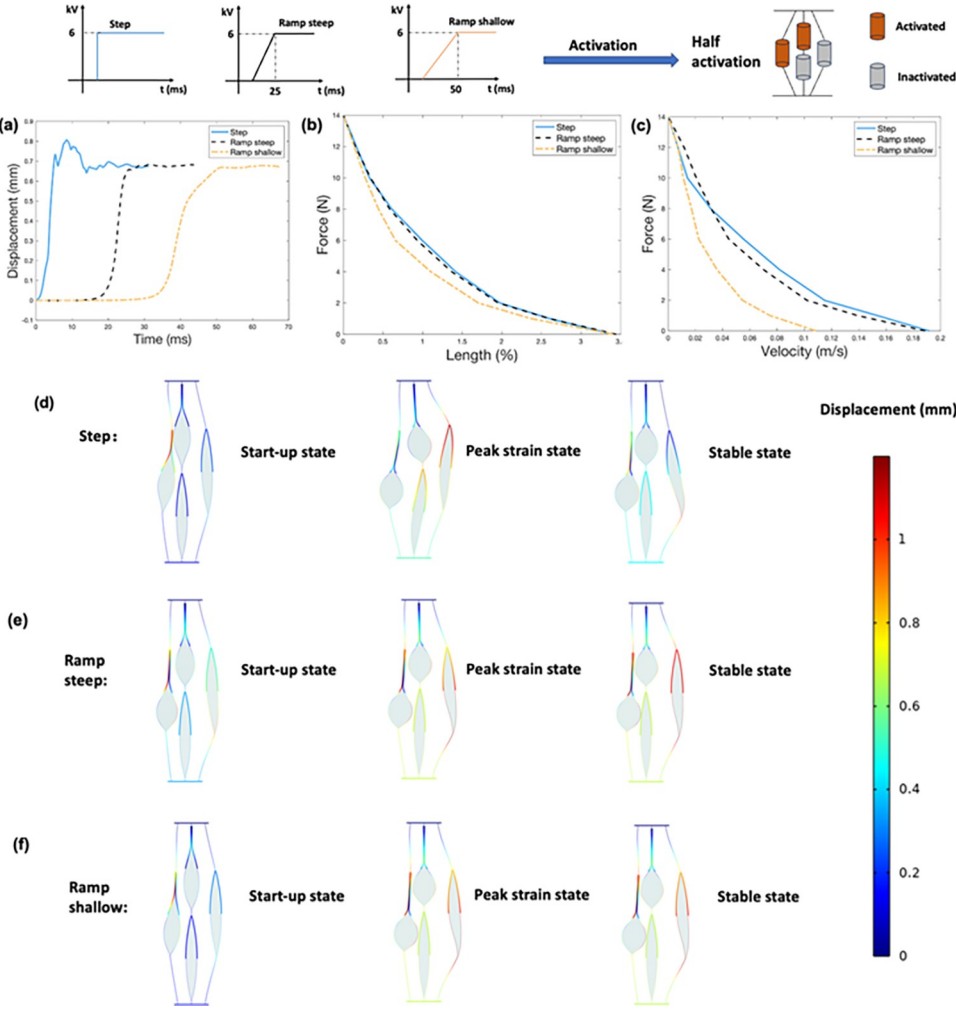

**Fig 6. Results obtained from the finite element model for profile of activation signals at 50% activation level.** (a). Displacement-time response for three configurations under no-load condition. (b). Force-length (i.e., maximum contractile strain under applied loads) relationship (c). Force-velocity relationship. (d). Three states of the step input under no load conditions. (e). Three states of the steep ramp input under no load conditions. (f). Three states of the shallow ramp input under no load conditions. Note that activation displacements shown in d-f are 1:1 scale.

increase of 93% in contraction velocity. The decrease in contraction velocity for the shallow ramp signal compared to step signal may come from the time it takes to charge the terminal to the threshold voltage [15]. Fig 6D to 6F show three states of different profile configurations under no load conditions. The distortion of inactivated actuators shows no significant difference between the step and ramp activation signals.

## D. Frequency of the activation signal

Fig 7 shows the results obtained from the finite element model for frequency of activation signals at 50% activation level. Fig 7A and 7B show the displacement-time responses under no-load condition for frequencies ranging from 0.5 $Hz$ to 1000 $Hz$. The displacement patterns change with frequency. As the frequency increases, the total displacement decreases due to the Peano-HASEL actuator limitation in response time, it is unable to respond to high frequencies such as 1000 $Hz$. Fig 7C further shows the relationship between activation frequency and the dynamic and quasi-static components of the total displacement. As the activation frequency increases from 0.5 $Hz$ to 1000 $Hz$, the quasi-static components of displacement increase from 0.35 $mm$ to 0.47$mm$, while the dynamic components decrease from 0.35 $mm$ to 0.01 $mm$. The dynamic component at 1000 $Hz$ is almost negligible (0.01 $mm$), and its quasi-static component of 0.47$mm$ is 33% smaller than the maximum displacement of 0.7 $mm$ at lower frequencies (less than 40 $Hz$). Fig 7D and 7E shows the force-length (i.e., maximum contractile strain under applied loads) and force-velocity relationships of seven configurations, respectively. For a given load, periodic signal with lower frequency produces greater strain output but lower contraction velocity. Compared to 0.5 $Hz$ signal, 1000 $Hz$ signal results in an average reduction of 128% in strain output and an average increase of 325% in contraction velocity. The maximum output force of 0.5 $Hz$ is 3 times that of 1000 $Hz$. Fig 7F to 7H show two states of different frequency configurations under no load conditions. Unlike constant activation signals, periodic signals do not lead the system to a stable position. Therefore, the stable state of the system is not given in Fig 7F to 7H. The distortion of inactivated actuators shows no significant difference for signals of different frequencies.

## E. Phase of the activation signal

Fig 8 shows the results obtained from the finite element model for phase of activation signals at 50% activation level. Fig 8A shows the displacement-time responses under no-load condition of two configurations. The maximum displacement output of 90 degrees phase is greater than that of 0 degree. Fig 8B and 8C shows the force-length (i.e., maximum contractile strain under applied loads) and force-velocity relationships of three configurations, respectively. For a given load, 40 $Hz$ sinusoidal input with 90 degrees phase produces greater strain output but the same level of contraction velocity as 0 degree. Compared to 0 degree, 90 degree signal results in an average increase of 57% in strain output. Fig 8D to 8F show two states of different phase configurations under no load conditions. The distortion of inactivated actuators shows no significant difference for signals of different phases.

## E. Findings on activation strategies

Activation strategies for the PH artificial muscle have been investigated as a function of activation level, activated position, and activation signals. Different activation modes have a profound impact on the performance of the PH artificial muscle in terms of displacement output, force output, and contraction velocity. Table 2 summarizes the findings regarding activation strategies of the PH artificial muscle.

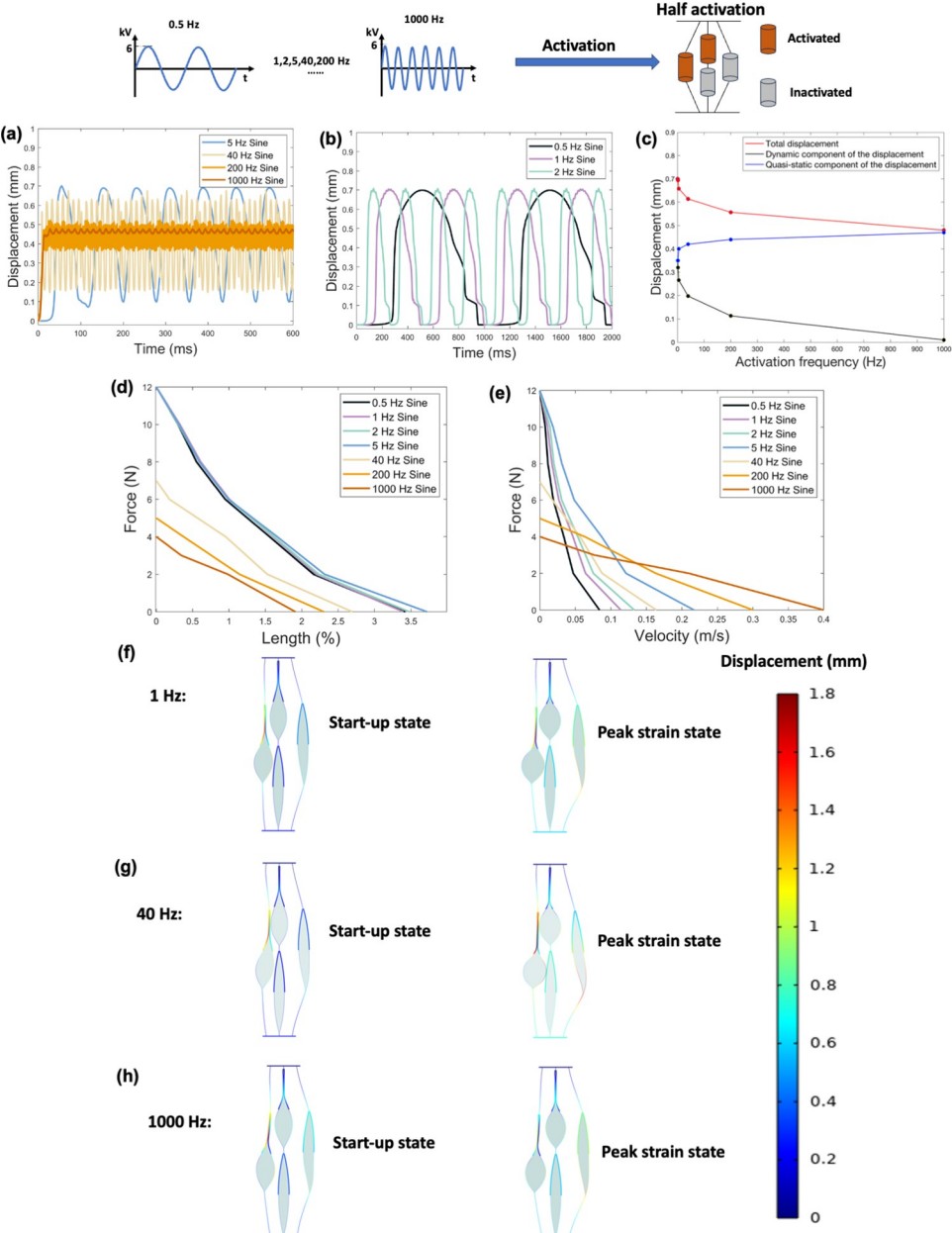

**Fig 7. Results obtained from the finite element model for frequency of activation signals at 50% activation level.**
(a). Displacement-time response for four configurations (i.e., 5 *Hz*, 40 *Hz*, 200 *Hz*, and 1000 *Hz*) under no-load
condition. (b). Displacement-time response for three configurations (i.e., 0.5 *Hz*, 1 *Hz*, and 2 *Hz*) under no-load
condition. (c). Displacement-activation frequency relationship. Dynamic and quasi-static components of the total
displacement were considered. (d). Force-length (i.e., maximum contractile strain under applied loads) relationship.
(e). Force-velocity relationship. (f). Two states of the 1 *Hz* sinusoidal input under no load conditions. (g). Two states of
the 40 *Hz* sinusoidal input under no load conditions. (h). Two states of the 1000 *Hz* sinusoidal input under no load
conditions. Note that activation displacements shown in f-h are 1:1 scale.

## Discussion

Inspired by the human neuromuscular system, this paper created a finite element model of a
PH artificial muscle and investigated how the activation strategies affect the performance of
the PH artificial muscle. This paper presents a pilot study on activation strategies for multi-

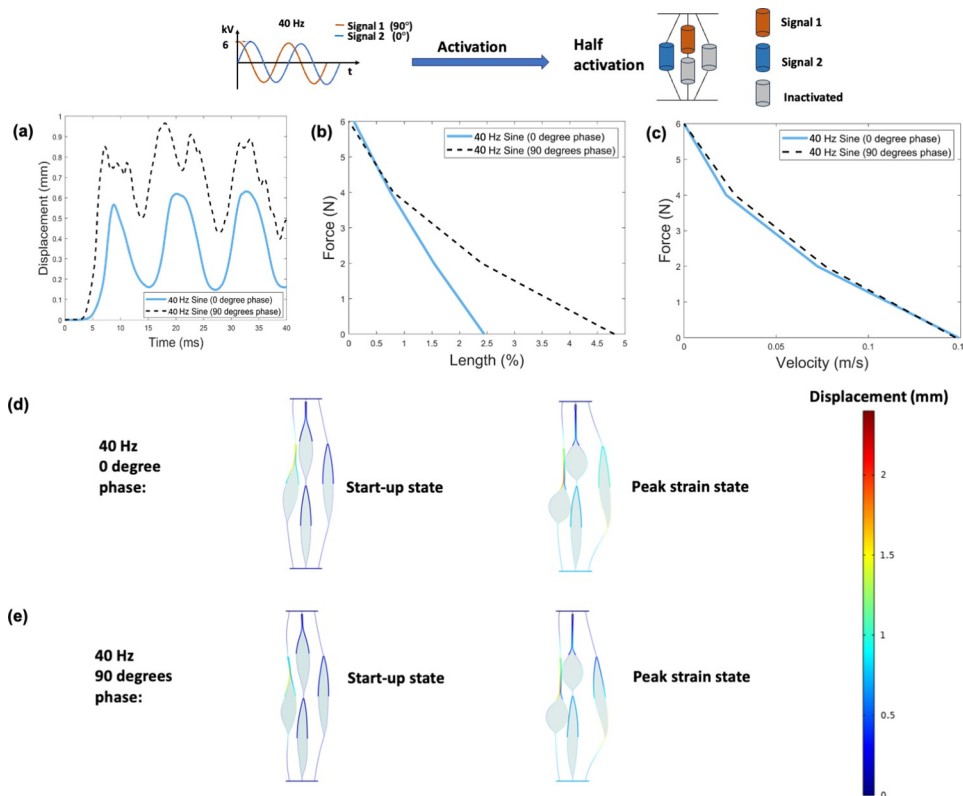

**Fig 8. Results obtained from the finite element model for phase of activation signals at 50% activation level.** (a). Displacement-time response for three configurations under no-load condition. (b). Force-length (i.e., maximum contractile strain under applied loads) relationship (c). Force-velocity relationship. (d). Two states of the 40 *Hz* sinusoidal input with a phase of 0 degree under no load conditions. (e). Two states of the 40 *Hz* sinusoidal input with a phase of 90 degree under no load conditions. Note that activation displacements shown in d-e are 1:1 scale.

**Table 2. Activation strategies for the Peano-HASEL artificial muscle.**

| Activation Modes | | Advantages | Disadvantages |
|---|---|---|---|
| Activation level | Low (e.g., 25%) | Low strain output, force output, and contraction velocity | High efficiency |
| | High (e.g.,100%) | High strain output, force output, and contraction velocity | Low efficiency |
| Activated position | Side | None compared to middle and mix | Low strain output, force output, and contraction velocity. High vibration. |
| | Middle | High strain output and contraction velocity, smooth displacement-time response | Low force output |
| | Mix | High force output | Low strain output and contraction velocity |
| Profile of activation signal | Step | High contraction velocity | High vibration |
| | Ramp | Smooth and overdamped displacement-time response | Low contraction velocity |
| Frequency of activation signal | Low (e.g., 5 *Hz*) | High strain output and force output | Low contraction velocity |
| | High (e.g., 1000 *Hz*) | High contraction velocity | The PH artificial muscle cannot respond quickly enough to the high frequency activation. |
| Phase of activation signal | Out of Phase (e.g., 90˚) | High strain output | None compared to in phase |
| | In phase | None compared to out of phase | Low strain output |

actuator-based artificial muscles, aiming to develop customized activation strategies for various applications. Based on the simulation results, greater strain output (106%) and greater contraction velocity (128%) were achieved by activating all the actuators, but overall energy efficiency was sacrificed by 47%. To improve efficiency, the activation level should be controlled when possible. Middle actuators should be activated prior to side actuators for smoother displacement (i.e., reduced vibration), greater force output (70%), greater strain output (130%), and greater contraction velocity (71%). The ramp activation signal with low frequencies (smaller than 5 $Hz$) is suitable for applications favouring controllable displacement, while the step activation signal with high frequencies (up to 1000 $Hz$) produces greater contraction velocity (325%). Besides, using signals with phase difference to activate different actuators can amplify the strain output by 57%.

Specifically, the more actuators that were activated, the greater the vibration, which was detrimental to control. Although the strain output and contraction velocity of 100% activation level were 106% and 128% greater than those of 25% activation level, considering that activating four actuators requires four times the energy input than activating one actuator, activating all the four actuators leads to a deduction of 47% in efficiency. there was a trade-off between system output and energy efficiency. For the position of activated actuators, symmetrically activating side actuators led to greater vibration than middle actuators. At 50% activation level, compared to activating two side actuators, activating two middle actuators resulted in an average increase of 130% and 71% in output strain and contraction velocity, respectively (likely to be due to the parallel and series configuration). Activating one side actuator and one middle actuator produced a force output that was 70% greater than activating two side actuators. As for the activation signals, ramp signals produced smooth and overdamped displacement-time response, which was beneficial for precise position control. The profile of the activation signal did not affect the strain output of the system. However, compared to shallow ramp signal, the step signal resulted in an average increase of 93% in contraction velocity, possibly due to the charging time required to reach voltage threshold [21]. The displacement patterns changed with frequency. As the frequency increased, the total displacement decreased due to that the Peano-HASEL actuator cannot respond at high frequencies such as 1000 $Hz$. Specifically, as the activation frequency increases, the quasi-static components of displacement increase, and the dynamic components decrease. This trend is consistent with the force-frequency characteristics of mammalian skeletal muscles [22,23], i.e., twitch contraction at low frequency and tetanic contraction at high frequency, indicating that the activation strategies for multi-actuator-based artificial muscles are somewhat similar to those of biological muscles. In addition, lower frequency periodic signals resulted in higher strain output but slower contraction velocity for a given load. When a 1000 $Hz$ signal was used instead of a 0.5 $Hz$ signal, the average strain output was reduced by 128% and the average contraction velocity was increased by 325%. Three times as much force can be produced at 0.5 $Hz$ as at 1000 $Hz$. Besides, for the phase of the signal, using sinusoidal signals with a 90-degree phase difference to activate two actuators resulted in an average increase of 57% in strain output compared to no difference.

In the human body, the force exerted by a muscle depends on which motor units are recruited and the rate at which they discharge action potentials [8,9]. Higher activation level (about 40%) in a micromechanical muscle model produced on average 93% greater force than lower activation level (about 20%) [10]. Similar to human muscle, the more actuators that are activated, the greater the force output of the Peano-HASEL artificial muscle. For a given load, the strain output and contraction velocity of the Peano-HASEL artificial muscle at 100% activation level were 106% and 128% greater than those of 25% activation level. The spatial distribution and location of the activated fibres within human muscle also affects the stress and strain distribution within the muscle [10]. Human muscles regulate the number and

combination of motor units activated for different applications [12]. For Peano-HASEL artificial muscle, activating the middle actuators facilitates strain output and contraction velocity, while activating side and middle actuators together is beneficial to force output. As for the discharging rates, faster motor units which respond to stimulation with higher frequency generate higher forces than slower motor units during contraction [24,25]. There was a significant correlation between muscle shortening strain rate and myoelectric frequency [26]. The exerted force of a single motor unit varied about 3 to 15 times for different discharging rates [9]. Compared to human muscles, the force output of Peano-HASEL artificial muscle varied by a factor of 3 for activation signals between 0.5 $Hz$ and 1000 $Hz$. Besides, compared to 0.5 $Hz$ signal, 1000 $Hz$ signal resulted in an average reduction of 128% in strain output and an average increase of 325% in contraction velocity.

The strain and force output of a single Peano-HASEL actuator is limited. One solution is to combine several actuators together to amplify the force and strain output [13]. A prosthetic finger was constructed by connecting 84 Peano-HASEL actuators in parallel and in series and was tested experimentally [7]. Similarly, 80 Peano-HASEL actuators were combined in the computational model to restore the force output of human triceps surae muscles [3]. Although the overall force output of the system is amplified, in the literature [7,10], all the actuators are activated with same shaped input signals and on-off control strategy (step input). The on and off activation signal has also been widely used in other electrically driven soft actuators [27,28]. While research in activation for an actuator has shown that optimizing the phase and frequency of the input could achieve precise position control and smooth force output [14]. The displacement patterns of PH artificial muscle changed with frequency. Constant activation signals produced a constant displacement to a fixed position. However, applications like human gait require precise position control and repeated contraction with a frequency of approximately 1 $Hz$ [29]. The PH artificial muscle shows the same trend that a low activation signal (lower than 5 $Hz$) is beneficial for applications favouring large and controllable movement. As the frequency increases, the amplitude of the displacement decreases due to that the Peano-HASEL actuator cannot respond at high frequencies such as 1000 $Hz$. Sinusoidal activation signals are also made from ramped profiles as opposed to on-off signals used in others' research. This smooths the transition leading to more controllable responses. Additionally, using sinusoidal signals with a 90-degree phase difference amplified the strain output by 57% compared in phase activation.

The present paper used a friction setup within adjacent actuators in the model. The condition was set to non-adhesive contact. Whereas the inside the liquid dielectric and deformable soft electrode may function as deformable substrate and introduce adhesive force, making the conditions complicated. Experiments need to be conducted in the future to determine whether the contact between adjacent actuators is friction-dominated or cohesion-dominated. Additionally, the model used in this study set the PH actuators to not be in contact initially, which would not be the optimal packing configuration for the practical manufacturing of PH artificial muscles. These friction and initial configuration assumptions are limitations in the model used in this study. Future work should be focused on experimental studies of PH artificial muscle configurations to fully validate the model configuration.

## Conclusion

In conclusion, this paper presents a pilot study on activation strategies for multi-actuator-based artificial muscles via finite element analysis and shows that activation strategies have a great influence on the force production, displacement, and velocity of PH artificial muscles. The simulation results indicated that middle actuators should be activated before side

actuators for smoother displacement, greater force output (70%), greater strain output (130%), and greater contraction velocity (71%). Greater strain output (106%) and contraction velocity (128%) were achieved by activating all the actuators. The ramp activation signal with low frequencies (smaller than 5 *Hz*) is suitable for applications favouring controllable displacement, while the step activation signal produces greater contraction velocity (325%). Furthermore, by activating actuators out of phase, the strain output can be increased by 57%. This paper begins the investigation into activation strategy for multi-actuator-based muscle with the goal of developing customized activation patterns for various applications. Future work is needed to investigate the full design space of customized activation strategies, which would enable the application of artificial muscle technologies in areas such as prosthetics and manufacturing.

## Supporting information

**S1 Fig.**
(TIF)

**S1 File.**
(DOCX)

## Author Contributions

**Conceptualization:** Andrew Weightman, Glen Cooper.

**Formal analysis:** Zhaozhen Liu.

**Investigation:** Zhaozhen Liu.

**Methodology:** Zhaozhen Liu, Andrew Weightman, Glen Cooper.

**Software:** Zhaozhen Liu.

**Supervision:** Andrew Weightman, Glen Cooper.

**Validation:** Harrison McAleese.

**Writing – original draft:** Zhaozhen Liu.

**Writing – review & editing:** Harrison McAleese, Andrew Weightman, Glen Cooper.

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
