## [Decision Letter · Decision Letter 0]

13 Nov 2024

PONE-D-24-45016Bioinspired activation strategies for Peano-HASEL artificial musclePLOS ONE

Dear Dr. Cooper,

Thank you for submitting your manuscript to PLOS ONE. After careful consideration, we feel that it has merit but does not fully meet PLOS ONE’s publication criteria as it currently stands. Therefore, we invite you to submit a newer version of the manuscript that addresses the points raised during the review process, through major revisions.

We look forward to receiving your revised manuscript.

Kind regards,

Massimo Mariello

Academic Editor

PLOS ONE

Reviewers' comments:

Reviewer's Responses to Questions

**Comments to the Author**

1. Is the manuscript technically sound, and do the data support the conclusions?

Reviewer #1: Yes

Reviewer #2: Yes

2. Has the statistical analysis been performed appropriately and rigorously? 

Reviewer #1: Yes

Reviewer #2: N/A

3. Have the authors made all data underlying the findings in their manuscript fully available?

Reviewer #1: Yes

Reviewer #2: Yes

4. Is the manuscript presented in an intelligible fashion and written in standard English?

Reviewer #1: Yes

Reviewer #2: Yes

5. Review Comments to the Author

Reviewer #1: This manuscript by Liu et al. discusses activation of skeletal muscle units by developing a finite element model (FEM) of HASEL artificial muscle. Effect of activation strategies on various responses such as displacement-time, force-length etc. are studied. Simulation results predict and narrow down activation strategies that impact the muscle function the most and has implications in developing artificial muscles in robotic and/or medical applications (prosthetics).

Overall, the efforts are commendable but the manuscript lacks underlying insights in a few places and just states the results verbatim. That should be addressed before it can be considered for publication.

1) The abstract starts from the lines “artificial muscle and investigated the effect of activation strategies…..” AND AFTER, should be made more generic.Please explain broadly what is activation and why strategies are needed in half or one line. Then dive deep into the intricacies. Right now, it is abruptly talking about different relationships without any context. Also numbers of as strain output and others, carry no meaning to a reader without any insight. What are middle actuators? From When is a contraction velocity greater? Please revise the entire abstract in a manner that a reader can understand the paper summary and key results instead of the specific jargon that it now consists of. Please keep in mind that your readers at Plos One will be varied as this journal strives (in their own words) for “an inclusive journal community working together to advance science by making all rigorous research accessible without barriers”.

2 ) Page 2 - “However, the force output of a single PH actuator is insufficient to meet the needs of many applications such as prosthetics………..” - Why insufficient? Please present a complete story instead of citing literature. Do the authors expect the readers to go back to cited articles every now and then?

3) Page 4, Lines 73-74 - Why is this work unique? What does it add to the domain? Please highlight the key novelty after describing the aim of this research. Not by saying “investigates activation stratgies”, but providing the readers the main insight and importance of the work.

4) Page 6, Validation - I did not understand how validation was done. It just cites Ref.3 and Ref.7. Can there be any figures or tables to compare the data??

5) Page 7, Fig2 - I do not see any explanations in the main writing about specific sub-figures (signals etc.). All sub-figures need to be mentioned and interpreted.

Reviewer #2: The authors have created a finite element model of an assembly of peano-HASEL actuators, and studied the model predictions of actuator force, strain, and velocity outputs in response to various bio-inspired activation strategies. The model is well-constructed. Results are presented clearly throughout the manuscript and summarized in a tabulated form towards the end. The benefits and/or drawbacks of the studied activation strategies are highlighted and discussed within the context of potential applications.

I have a few comments for minor revision:

1) For the force-length relationships that are shown, force is the load F that is applied as shown in Fig. 1, but the length term is not defined clearly. Is it the % change in length of the overall actuator assembly? If so, is it the maximum contractile strain that is reached after activating the actuators under varying applied load F? The authors should clarify this in text, perhaps in the Methods section where the model setup and post processing are described.

2) Regarding activation frequency, the authors state in the Results and the Discussion sections that the displacement amplitude decreases with increasing frequency (lines 272-274 and 352-354). While this is true overall, there is a noticeable difference between the amplitudes of the dynamic oscillations and the quasi-static tension that is built-up especially at high frequencies. For instance, at 1000 Hz, the oscillation amplitude is very small, at least an order of magnitude smaller than the oscillation amplitudes at lower frequencies (40 Hz and below), but the total quasi-static displacement is around 0.5 mm at 1000 Hz (Fig. 7a) which is still comparable to the maximum displacement of about 0.7 mm. Interestingly, this is reminiscent of the classical force-frequency relationship of biological skeletal muscle, which shows dynamic twitching at low frequency and tetanic contraction at high frequency. The authors should highlight these differences in the relationship between activation frequency and the dynamic vs. quasi-static components of displacement.

3) In the Results section, in the paragraph describing the results shown in Fig. 5, the authors state (line 234): “The maximum force output of the mix configuration is 114% of that of side configuration”. However, Fig. 5b shows the opposite, showing a maximum force of about 16 N for the side configuration and about 14 N for the middle configuration. I.e., the maximum force output of the side configuration is 114% that of the middle configuration.

4) Fig. 1 caption (line 86): “…arrange of constant loads…” should be revised to “…a range of constant loads…”

5) Materials and Methods section B and Fig. 2 caption (lines 161, 164, 172, and 173): The word “activate” is repeated in the phrase “…used to activate the activate the actuators…”

6) In Fig. 3, the last two rows are both labeled “(f)” instead of “(f)” and “(g)”. Also, the last sentence in the figure caption (line 204) reads “… shown in d-f are…” instead of “…shown in d-g are…”

6. PLOS authors have the option to publish the peer review history of their article (what does this mean?). If published, this will include your full peer review and any attached files.

Reviewer #1: No

Reviewer #2: No

---

## [Author Response · Author response to Decision Letter 0]

8 Jan 2025

[Please note that a reviewer response document has been uploaded - reviewers may find it easier to look at this document where changes have been marked in yellow to view the authors responses]

Thank you for giving us the opportunity to submit a revised draft of our manuscript titled ‘Bioinspired activation strategies for Peano-HASEL artificial muscle’ to PLOS ONE 

We appreciate the time and effort that you and the reviewers have dedicated to providing your valuable feedback and comments on our manuscript. We have been able to incorporate changes to reflect suggestions provided by the reviewers. Changes are marked in yellow. In addition, we have reformatted the manuscript according to formatting guidelines.

Here is a point-by-point response to the reviewer’ s comments.

Comments from the Reviewer #1: 

Summary of Comments: 

This manuscript by Liu et al. discusses activation of skeletal muscle units by developing a finite element model (FEM) of HASEL artificial muscle. Effect of activation strategies on various responses such as displacement-time, force-length etc. are studied. Simulation results predict and narrow down activation strategies that impact the muscle function the most and has implications in developing artificial muscles in robotic and/or medical applications (prosthetics).

Overall, the efforts are commendable, but the manuscript lacks underlying insights in a few places and just states the results verbatim. That should be addressed before it can be considered for publication.

Response:

Thank you for taking the time to read our manuscript. We really appreciate the feedback and insightful comments on our manuscript. Several changes were made based on your feedback, which we hope will improve the clarity and quality of the manuscript and made it more accessible to readers from various backgrounds. Please see responses to specific comments below for details.

Please see specific comments below:

Specific Comments:

Comment 1:

The abstract starts from the lines “artificial muscle and investigated the effect of activation strategies….” AND AFTER, should be made more generic. Please explain broadly what is activation and why strategies are needed in half or one line. Then dive deep into the intricacies. Right now, it is abruptly talking about different relationships without any context. Also numbers of as strain output and others, carry no meaning to a reader without any insight. What are middle actuators? From When is a contraction velocity greater? Please revise the entire abstract in a manner that a reader can understand the paper summary and key results instead of the specific jargon that it now consists of. Please keep in mind that your readers at Plos One will be varied as this journal strives (in their own words) for “an inclusive journal community working together to advance science by making all rigorous research accessible without barriers”.

Response 1:

Thank you for pointing out that our abstract contains too many technical terms and jargon which would make our abstract less accessible to the readers of the journal. We appreciate that the abstract should be more generic and understandable to readers from various backgrounds. We have revised the abstract to provide more generic insights and explain some terms used. 

Specifically, you asked us to explain:

• What activation is and why strategies would be needed: 

In artificial muscle, activation is the contraction of a single actuator or multiple actuators within the muscle. The number of activated actuators, timing and magnitude of activation (the activation strategy) will enable modulation of the artificial muscles force, displacement and contraction velocity. These activation strategies will mean that an artificial muscle will be able to change its performance without altering its hardware – making it more versatile in a range of applications or tasks.

• What is strain and the numbers that are quantified: 

Strain output referred to the displacement change which we agree was not clear and we had not been consistent in our terminology - apologies. We have modified the abstract to refer to displacement and so the numbers (percentages) now refer to displacement changes. Additionally, we have also defined “length” a term which is used commonly in human muscle biomechanics, as the maximal contractile strain.

• What are middle actuators: 

This study developed a finite element model of an artificial muscle consisting of four Peano-HASEL actuators arranged in three parallel groups in a diamond pattern (two actuators in series in the middle with one actuator in parallel either side). So the middle actuators refer to the two actuators in series in the middle of the artificial muscle. 

• From when is a contraction velocity greater:

Thank you for highlighting that the contraction velocity is not constant throughout the contraction. Again, we apologise for not being clear in our manuscript on this. The contraction velocities are greater in the initial phase of contraction, but the figures quoted are for the average velocity.

Below we have tried to incorporate these explanations and made corrections in the abstract

New Abstract:

Background

Human muscles perform many functions during activities of daily living, producing a wide range of force outputs, displacements, and velocities. This versatile ability is believed to be associated with muscle activation strategies, such as the number and position of activated motor units within the muscle, as well as the frequency, magnitude and shape of the activation signal. Activation strategies similar to those in the human neuromuscular system could increase the functionality of artificial muscles. Activation in an artificial muscle is the contraction of a single actuator or multiple actuators within the muscle. The number of activated actuators, timing and magnitude of activation (the activation strategy) will enable modulation of the artificial muscles force, displacement and contraction velocity. These activation strategies will mean that an artificial muscle will be able to change its performance to modulate its displacement, length (maximal contractile strain) and velocity for various loading conditions without altering its hardware – making it more versatile in a range of applications or tasks.

This study aims to investigate the effect of activation strategies on the displacement-time response, force-length relationship, and force-velocity relationship of a Peano-hydraulically amplified self-healing electrostatic (HASEL) artificial muscle.

Method 

This study developed a finite element model of an artificial muscle consisting of four Peano-HASEL actuators arranged in three parallel groups in a diamond pattern (two actuators in series in the middle – middle actuators, with one actuator in parallel either side – side actuators). Bioinspired activation strategies were applied to the artificial muscle. Specifically, the number of activated actuators (i.e., activation level), the position of activated actuators, the profile, frequency, and phase of the activation signal were investigated.

Results 

Activating more actuators resulted in increased displacement (106%) and increased average contraction velocity (128%), but overall energy efficiency was sacrificed by 47%. The distortion of inactivated actuators was mitigated by symmetric and phased activation. Phased activation refers to activating middle actuators before side actuators. In addition, displacement patterns of the Peano-HASEL artificial muscle changed with activation signal frequency. The ramp activation signal with low frequencies (less than 5 Hz) is suitable for applications favouring controllable displacement, while the step activation signal produces greater average contraction velocity (325%) which would be advantageous for applications requiring a fast response.

Conclusion

This paper demonstrates that activation strategies can enhance multi-actuator artificial muscle function without changing the physical hardware configuration. Specifically, activation strategy can, improve displacement control, contraction velocity and output force. Future work should focus on more complex artificial muscle arrangements and test activation strategies in practical experiments.

Comment 2:

Page 2 - “However, the force output of a single PH actuator is insufficient to meet the needs of many applications such as prosthetics……….” - Why insufficient? Please present a complete story instead of citing literature. Do the authors expect the readers to go back to cited articles every now and then?

Response 2:

Thank you for your suggestion. On Page 3, Lines 53 to 58, we have rewritten these sentences to explain why the force output of one actuator is insufficient and complete the story. We used prosthetics as an example, the force output of one actuator is about 15 to 40 N, whereas a human muscle of the same length can produce forces greater than 200 N. So, the force output of one actuator is not enough. And a commonly used solution to this problem is to stack multiple actuators in parallel. 

Please see the sentences below for changes:

On Page 3, Lines 53 to 58:

‘’However, the force output of a single PH actuator is insufficient to meet the needs of many applications such as prosthetics. For instance, the maximum force output of a single PH actuator with commonly used geometry ranges from 15 to 40 N [2],[3], whereas a human skeletal muscle of similar geometry produces forces greater than 200 N [5]. Therefore, multiple PH actuators must be stacked to achieve force amplification. This stacked parallel configuration has been used to create linear grippers [6] and high-speed prosthetic fingers [7].”

Comment 3:

Page 4, Lines 73-74 - Why is this work unique? What does it add to the domain? Please highlight the key novelty after describing the aim of this research. Not by saying “investigates activation strategies” but providing the readers the main insight and importance of the work.

Response 3:

Thank you for your suggestion. On Page 4, Lines 82 to 91, we have added several sentences after the aim of this study to summarize the key novelty and help readers understand the importance of the work.

Please see the sentences below for changes:

On Page 4, Lines 82 to 91:

“This paper aims to investigate how the activation strategies of PH actuator based artificial muscle (i.e., groups of PH actuators) affect its force output and length, as well force-velocity relationships. Previously researchers have focussed on the physical hardware of artificial muscle design [2],[3]. We are investigating their control through activation strategy which may enable additional functional modulation without hardware changes. This could both improve performance but also make artificial muscles more versatile to perform in a range of applications. Other researchers have looked at activation signals but mainly on ramp and magnitude variation [3],[6]. Here we present a biomimetic approach to explore the design space in more aspects which we hypothesise will unlock greater functionality. Specifically, activation strategies will be investigated in three aspects: number of activated actuators, position of activated actuators, and activation signals. For the activation signals, the profile, phase, and frequency of the signals will be studied.’’

Comment 4:

Page 6, Validation - I did not understand how validation was done. It just cites Ref.3 and Ref.7. Can there be any figures or tables to compare the data??

Response 4:

Thank you for your suggestion. We performed experiments for one and two actuators and compared the experimental results with FE simulations to validate the FE model. The main difference between the model used in the present study (i.e., four actuators model) and our previous models (i.e., one and two actuators’ models) is the number of actuators used. All the other factors like geometric parameters, material properties, the way to activate the actuators, and the way to connect actuators are the same. We have then made the assumption that as these other configurations based on the same parameters show good agreement with other data that it gives strong evidence that our model is accurate. 

We agree that validation plots from the one or two actuators’ models would help readers understand the validation process. Hence, on Page 8, Lines 159 to 169, we have clarified how we have approached validation. In addition, we provided the plots for the validation of one and two actuators’ models in the supplementary material. In the discussion section, we have also suggested that future work should be focused on experimental studies of our modelled configurations.

Please see the sentences below for changes:

On Page 8, Lines 159 to 169:

“Validation: The actuators within the finite element model were compared with previously published results of both an analytical model and an experiment [3] and found to be in close agreement with errors of less than 5%. Additionally in our previous work [17] we compared the finite element model results to experimental results of a bipennate PH actuator artificial muscle (i.e., two PH actuators in a symmetric angled arrangement). This again had low errors of less than 10% across length changes of 1-6%. These finite element models were then extended to create an artificial muscle of four PH actuators in a diamond arrangement, as shown in Fig 1. The diagrams of actuator arrangements and their force-length characteristics compared to experimental and analytical data from the literature can be found in the supplement material S1 Fig.”

On Page 24, Lines 482 to 483:

‘‘These friction and initial configuration assumptions are limitations in the model used in this study. Future work should be focused on experimental studies of PH artificial muscle configurations in order to fully validate the model configuration.’’

Supplementary material:

“S1 Fig shows the finite element models of one and two actuators and their force-length characteristics compared to experimental and analytical data from the literature. A boundary load P is applied at the bottom end of the actuator via point A and corresponding displacements are recorded. The force-strain relationships obtained from the finite element models are then compared to experimental data. The error between finite element simulations and experiments is less than 10%. This is strong evidence that the finite element simulations provide an accurate representation for PH actuator artificial muscle arrangements.

The main difference between the model used in the present study (i.e., four actuators) and our previous models (i.e., one and two actuators) is the number of actuators used. All the other factors like geometric parameters, material properties, the way to activate the actuators, and the way to connect actuators are the same. We have then made the assumption that as these other configurations based on the same parameters show good agreement with other data that it gives strong evidence that our model is accurate. Future work should begin experiments based on the promising activation strategies discovered in this paper.

[Additional figure S1 Fig added - see downloaded files]

Comment 5:

Page 7, Fig2 - I do not see any explanations in the main writing about specific sub-figures (signals etc.). All sub-figures need to be mentioned and interpreted.

Response 5:

Thank you for your suggestion, which highlighted our omission to describe this fully. Hence, to improve the clarity, we reproduced the Fig 2 to add labels (a), (b), and (c). On Page 10, Lines 195 to 209, the interpretation in the main text was also revised.

Please see the revised Fig and sentences below for changes:

Fig 2. Activation strategy investigation for the PH artificial muscle consisting of four actuators. (a). The number of activated actuators was studied. One to four actuators were activated, corresponding to 25% to 100% activation level. (b). The number of activated actuators was studied. For 50% and 75% activation level, the middle and side actuators were activated in different combinations to study the effect of activation position. (c). For the activation signal, (1) the profile, (2) frequency, and (3) phase of the 

---

## [Editor Report · Decision Letter 1]

21 Jan 2025

Bioinspired activation strategies for Peano-HASEL artificial muscle

PONE-D-24-45016R1

Dear Dr. Cooper,

We’re pleased to inform you that your manuscript has been judged scientifically suitable for publication and will be formally accepted for publication once it meets all outstanding technical requirements.

Kind regards,

Massimo Mariello

Academic Editor

PLOS ONE
---

## [Editor Report · Acceptance letter]

25 Jan 2025

PONE-D-24-45016R1 

PLOS ONE

Dear Dr. Cooper, 

I'm pleased to inform you that your manuscript has been deemed suitable for publication in PLOS ONE. Congratulations! Your manuscript is now being handed over to our production team.

Kind regards, 

on behalf of

Dr. Massimo Mariello 

Academic Editor

PLOS ONE
